# *Leptospira* Seroprevalence in Free-Ranging Long-Tailed Macaques (*Macaca fascicularis*) at Kosumpee Forest Park, Maha Sarakham, Thailand

Natapol Pumipuntu [1,2,3,*], Tawatchai Tanee [1,4], Pensri Kyes [5], Penkhae Thamsenanupap [1,4], Apichat Karaket [6] and Randall C. Kyes [7]

1    One Health Research Unit, Mahasarakham University, Maha Sarakham 44000, Thailand
2    Veterinary Infectious Disease Research Unit, Mahasarakham University, Maha Sarakham 44000, Thailand
3    Faculty of Veterinary Sciences, Mahasarakham University, Maha Sarakham 44000, Thailand
4    Faculty of Environment and Resource Studies, Mahasarakham University, Maha Sarakham 44150, Thailand
5    Department of Psychology, Center for Global Field Study, and Washington National Primate Research Center, University of Washington, Seattle, WA 98195, USA
6    Department of National Parks, Wildlife and Plant Conservation, Bangkok 10900, Thailand
7    Departments of Psychology, Global Health, and Anthropology, Center for Global Field Study, and Washington National Primate Research Center, University of Washington, Seattle, WA 98195, USA
*    Correspondence: natapol.p@msu.ac.th

**Abstract:** Background: Leptospirosis is a zoonotic disease that is ubiquitously distributed and is classified as a re-emerging infectious disease in humans and animals. Many serovars are carried by wildlife; all of them are capable of causing illness in humans. The purpose of this study was to investigate the prevalence of Leptospirosis in wild long-tailed macaques (*Macaca fascicularis*) at Kosumpee Forest Park, Mahasarakham, Thailand. Methods: A cross-sectional study was conducted at the park. Blood samples were collected via saphenous vein from 30 free-ranging long-tailed macaques. All samples were tested by the microscopic agglutination test. The *LipL32* gene was used to detect pathogenic *Leptospira* in blood samples by conventional polymerase chain reaction. Results: Screening of the 30 wild macaques showed an overall *Leptospira* seroreactivity of 13.33%. Three of 30 macaques reacted against *Leptospira* serovar Shermani and one macaque was infected with *Leptospira* serovar Sejroe. None of the macaques presented clinical signs of leptospirosis. None of the blood samples showed the detection of the *LipL32* gene. Conclusions: The results indicate that the long-tailed macaques at Kosumpee Forest Park may act as natural reservoirs for *Leptospirosis*. Further, the results provide evidence-based information indicating that several pathogenic *Leptospira* serovars are circulating in the wild macaques in the study area.

**Keywords:** leptospirosis; long-tailed macaques; MAT; seroprevalence; wildlife; zoonosis





## 1. Introduction

Leptospirosis is known as a major endemic bacterial zoonotic disease in Thailand. It is considered to be a neglected tropical disease in many countries, especially in tropical areas [1]. The zoonosis is caused by Gram-negative spirochetes, a spiral-shaped pathogenic bacterium called *Leptospira interrogans*. The pathogenic *Leptospira* bacteria has more than 300 diverse serovars [2]. A wide variety mammalian species can play a role as disease reservoirs and carriers that transmit the pathogen to humans and other animals. They harbor the pathogenic *Leptospira* spp. in renal tubules. These bacteria are shed from the animal reservoirs through their urine and contaminate the environment, usually soils and water [3]. Humans and animals may then be exposed to these pathogenic *Leptospira* spp. via direct contact with infected animals or through indirect contact with the contaminated water [4].

Detection of antibody titer against *Leptospira* spp. by the microscopic agglutination test (MAT) is the gold standard method used mainly for serodiagnosis and seroprevalence of leptospirosis in both humans and animals. Given its ability to diagnose a specificity for serogroups and serovars, this method can identify the specific circulating serovars of infected human and animal hosts from their regions [5]. In Thailand, previous reports have revealed that many pets and livestock species serve as reservoir hosts for several predominant serovars of the pathogenic *Leptospira interrogans*, including serovars *Canicola* and *Bataviae* [6,7] in dogs, serovars *Bratislava* and *Pomona* in pigs [8], serovar *Sejroe and Ranarum* in cattle [6,9], serovars *Ranarum* and *Shermani* in the bullfighting cattle [10], serovar *Pyrogenes* in rodents [11], and serovars *Mini, Shermani,* and *Ranarum* in sheep and goats [9]. Compared to the work with livestock, there is very little information regarding the status of leptospirosis in wildlife of Thailand, especially in long-tailed macaques (*Macaca fascicularis*) that often live in shared environments with humans and are considered the most frequently encountered non-human primate species in this region [12]. As such, these macaques represent a potential reservoir host for this disease and, in turn, transmit the pathogen to humans and other animals. Given the potential One Health concern, the purpose of the current study was to assess the prevalence of the *Leptospira* spp. serovar in a free-ranging population of long-tailed macaques at Kosumpee Forest Park (KFP), Maha Sarakham, Northeastern Thailand. This study was part of a larger, ongoing project addressing human–primate conflict and coexistence [13–15] and screening for macaque health [16,17] at KFP.

## 2. Materials and Methods

### 2.1. Ethics Statement

This project was conducted in accordance with the animal-use protocol approved by the Institutional Animal Care and Use Committee at Mahasarakham University (IACUC-MSU) for animal subjects research (protocol approval No. 0009/2016). All animal sample collection complied with the applicable laws of Thailand.

### 2.2. Sample Collection and Procedures

As detailed in our previous studies [15,17], KFP supports a free-ranging population of long-tailed macaques (approximately 850 macaques at the time of the study), distributed among five social groups [14]. KFP is located in Kosum Phisai district, about 27 km from Muang district (city area), Maha Sarakham, Northeast Thailand (Figure 1). There is extensive interaction between the macaques and the local residents and tourists, often resulting in various levels of conflict, including crop raiding, damage to buildings, and aggressive interactions.

Blood samples were collected (10–11 November 2018) from 30 randomly selected free-ranging macaques living around the KFP area during trapping for health screening. All trapping and sampling procedures were conducted by wildlife veterinary specialists from Mahasarakham University (MSU) following the protocol approved by the MSU Institutional Animal Care and Use Committee and the Thai Department of National Parks. Capture procedures, previously described by Pumipuntu and team [17], involved the use of a soft nylon mesh cage with a wooden frame, baited with groundnuts and bananas. An intramuscular injection of Tiletamine-zolazepam (Zoletil® 100 mg/mL, Virbac, Carros, France) was administered via 5 mL anesthetic blowpipes to sedate the macaques in the cage. Following anesthesia, the macaques were removed from the cage, and vital signs were monitored prior to sample collection. Blood samples (5 mL in volume) were aseptically collected via the saphenous vein from each macaque. The samples were then transferred into a sterile plain blood collection tube/red cap (BD Vacutainer, BD, Franklin Lakes, NJ, USA), placed at room temperature for 10 min and separated a serum by centrifugation at 3000 g for 15 min. The separated serum samples were transferred into 1.5 mL sterile microtube (Eppendorf, Hamburg, Germany). All blood samples were processed for DNA extraction with the QIAamp DNA mini kit (Qiagen, Valencia, CA, USA) according to

the manufacturer's instructions. All serum and DNA extraction samples were stored at −20 °C at the Veterinary Public Health Laboratory of the Faculty of Veterinary Sciences at Mahasarakham University for further assessment.

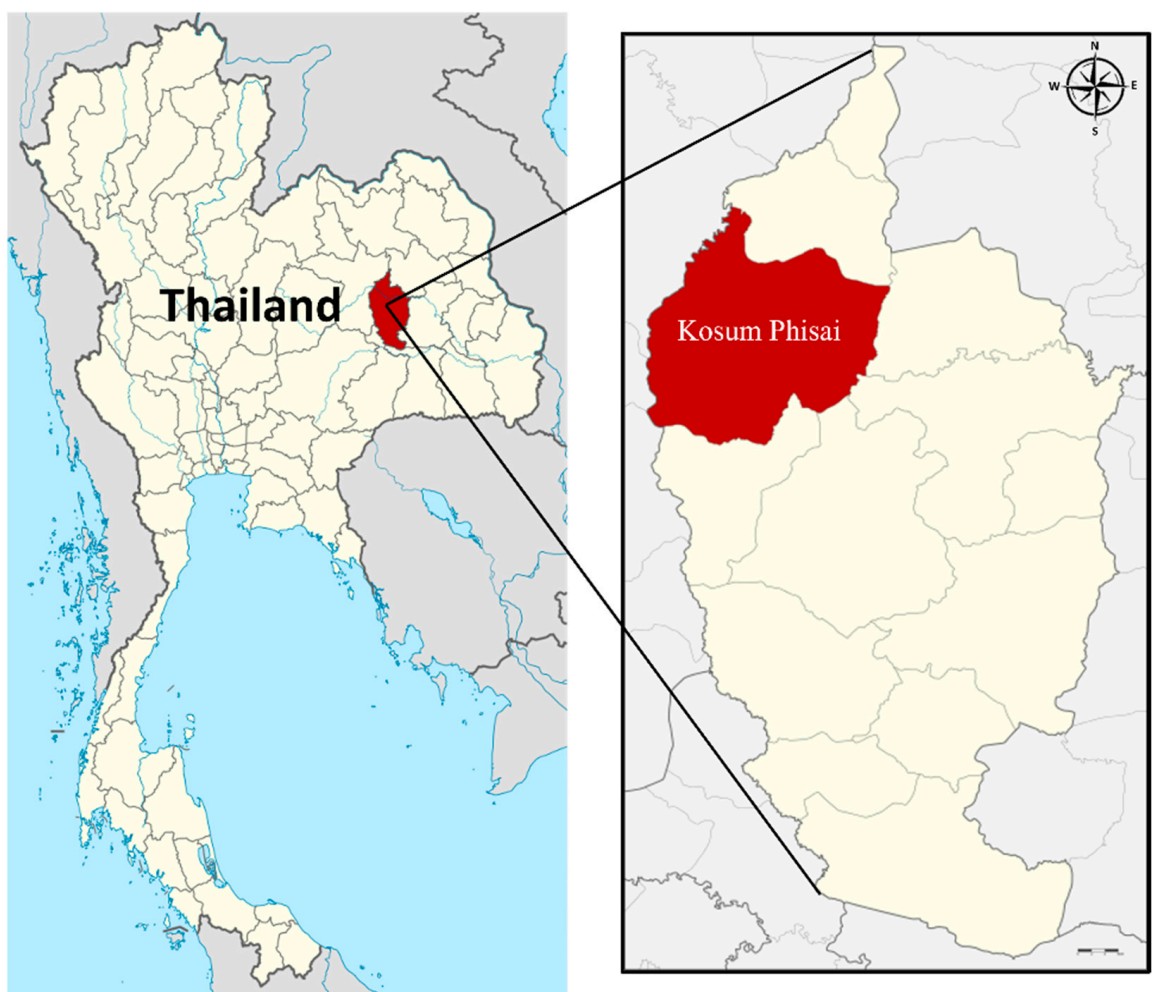

**Figure 1.** Map shows Kosum Phisai district, Maha Sarakham in the northeast of Thailand, where the Kosumpee Forest Park is located. (Source: https://en.wikipedia.org/wiki/Maha_Sarakham_province, accessed on 18 September 2022).

### 2.3. Microscopic Agglutination Test

All macaque serum samples were analyzed for the presence of *Leptosipira*-serovar-specific antibodies by the microscopic agglutination test (MAT) and performed according to standard methodology [18] by using a panel of 24 serovars that commonly circulate in the region, namely Pomona, Pyrogenes, Ranarum, Sarmin, Sejroe, Shermani, Tarassovi, Patoc, Canicola, Celledoni, Cynopteri, Ballum, Bataviae, Djasiman, Javanica, Louisiana, Manhao, Mini, Grippotyphosa, Australis, Autumnalis, Hebdomadis, Icterohaemorrhagiae, and Panama. All serum samples were diluted into 1:50 with phosphate-buffered solution (PBS) in a microtiter plate. Subsequently, they were serially diluted two-fold with PBS to set the dilutions to 1:100 to 1:1600. A dilution titer at 1:100 sera was considered as the level to screen the serum for positive leptospirosis.

### 2.4. Molecular Identification

Bacterial DNA was extracted from all blood samples using a QIAamp DNA mini kit (Qiagen, Hilden, Germany) according to the manufacturer's instructions. The concentration

and purity of DNA extractions from macaques' blood samples were assessed with the NanoDrop 1000 Spectrophotometer (Thermo Scientific, Branchburg, NJ, USA) by measuring the wavelength at OD 260 and 280 nm, and the purity was measured by obtaining the 260/280 ratio. These DNA extractions were stored at $-20\,^{\circ}$C for further molecular analysis. All DNA samples were assessed for the *LipL32* gene as a means of detecting pathogenic *Leptospira* bacteria.

Each PCR reaction was carried out in 25 μL, including the DNA template, Taq DNA Polymerase Master Mix (KAPA2G™ Robust HotStart ReadyMix PCR Kit, Kapa Biosystems, Wilmington, MA, USA), sterile deionized water, and 10 μmol/L of both forward and reverse primers of the *LipL32* gene. Forward and reverse primers used for targeting the *LipL32* gene were 45F (5′-AAG CAT TAC CGC TTG TGG TG-3′) and 286R (5′-GAA CTC CCA TTT CAG CGA TT-3′), respectively. These primers generated a fragment size of 242 base pairs of PCR product, which has been noted to provide both 100% sensitivity and specificity [19]. PCR amplification was conducted with one cycle of initial denaturation at 94 $^{\circ}$C for 4 min. PCR amplification was then performed following 35 cycles of denaturation at 94 $^{\circ}$C for 30 s, annealing to 53 $^{\circ}$C for 30 s, extension at 72 $^{\circ}$C for 1 min, and the final extension cycle at 72 $^{\circ}$C for 5 min. PCR products underwent electrophoresis by 2% agarose gels in 1X TAE buffer that was stained with ViSafe Red Gel Stain (Vivantis, Selangor, Malaysia) at 100 V for 30 min, and visualized under ultraviolet light. The fragments were identified via VC100bp Plus DNA ladder (Vivantis, Selangor, Malaysia). Sterile water was used as a negative control, and DNA of *Leptospira interrogans* serovar *Sejroe* was used as a positive control in each running of the gel electrophoresis.

### 2.5. Statistical Analysis

Descriptive statistics were used to determine the prevalence of leptospirosis infection and the presence of *Leptospira* serovar antibodies. Their seroprevalence (percentage) with binomial exact calculation and 95% confidence intervals (CI) were calculated via a free online program (http://sampsize.sourceforge.net/iface/index.html, accessed on 7 August 2022).

### 3. Results

Of the 30 macaques sampled at KFP, none of the animals presented clinical signs of leptospirosis. From the MAT analysis, antibodies against *Leptospira* spp. were detected in four (13.33%, 95% CI: 3.76–30.72%) of the macaques. Of the four positive animals, three reacted against *Leptospira* serovar Shermani (10%; 95% CI: 2.11–26.53%), and one reacted against *Leptospira* serovar Sejroe (3.33%; 95% CI: 0.08-17.22%) as shown in Table 1. However, of the 30 blood samples analyzed by PCR, none were found to possess the *LipL32* gene, which is used to indicate the presence of pathogenic *Leptospira* bacteria.

**Table 1.** MAT and PCR results of *Leptospira* spp. infection of wild long-tailed macaques.

| Monkey ID | MAT Result | *Leptospira* Serovar | *LipL32* Gene Detection |
|:---:|:---:|:---:|:---:|
| M 1 | - | | - |
| M 2 | - | | - |
| M 3 | - | | - |
| M 4 | - | | - |
| M 5 | - | | - |
| M 6 | - | | - |
| M 7 | 1:100 | Shermani | - |
| M 8 | - | | - |
| M 9 | - | | - |
| M 10 | - | | - |
| M 11 | - | | - |
| M 12 | - | | - |

**Table 1.** *Cont.*

| Monkey ID | MAT Result | *Leptospira* Serovar | *LipL32* Gene Detection |
|-----------|------------|----------------------|-------------------------|
| M 13 | - | | - |
| M 14 | - | | - |
| M 15 | - | | - |
| M 16 | - | | - |
| M 17 | - | | - |
| M 18 | - | | - |
| M 19 | - | | - |
| M 20 | 1:100 | Shermani | - |
| M 21 | 1:100 | Shermani | - |
| M 22 | - | | - |
| M 23 | - | | - |
| M 24 | 1:100 | Sejroe | - |
| M 25 | - | | - |
| M 26 | - | | - |
| M 27 | - | | - |
| M 28 | - | | - |
| M 29 | - | | - |
| M 30 | - | | - |

## 4. Discussion

Based on the detection of antibodies against *Leptospira* spp. that were found in the long-tailed macaques at Kosumpee Forest Park, the possibility exists that these macaques act as natural reservoir hosts for leptospirosis. Given the close contact between the macaques and local resident in this shared environment, the opportunity for cross-transmission of zoonotic pathogens presents a real concern; one that also has been noted by others [20,21]. The occurrence of *Leptospira* seropositivity from the long-tailed macaques in this study (13.33%) was lower than what has been found in some recent studies from other countries, including Southern, Central, and Eastern Thailand, with 48 out of 223 (21.52%) long-tailed macaques [21]; 8 out of the 12 monkeys (66.66%) from Sarawak, Malaysia [22]; 39 out of 83 wild African green monkeys (47%) on the Caribbean island of Saint Kitts, India [23]; 16 out of 52 capuchin monkeys (30.77%) from Colombia [24]; and 39 out of 50 tufted capuchin monkeys (78%) from Southeast Sao Paulo state, Brazil [25].

Regarding the MAT result, *Leptospira* serovar Shermani was the most prevalent serovar followed by *Leptospira* serovar Sejroe. This finding is concordant with a previous study that revealed that the most general serovars in humans and livestock in Thailand were Shermani, Ranarum, and Sejroe [26]. *Leptospira* serovars Ranarum and Shermani were also be found in long-tailed macaques inhabiting Southern, Central, and Eastern regions of Thailand, as described in a recent study [21]. Additionally, *Leptospira* serovar Sejroe was reported to be circulating in cattle in Maha Sarakham province [6], which represents the same general study area as this current study. Our research provides significant baseline epidemiological information that could be correlated with the occurrence of leptospirosis in both humans and animals in this study area. Moreover, this infectious disease is considered endemic and a significant public health concern in Maha Sarakham province. In fact, between 2004 and 2014, Kosum Phisai district (where our study site was located) reported the second highest number of human cases in the province [27]. Thus, there may be a possibility that various animal species, including wild macaques, could play a potential role in leptospirosis animal reservoirs at this study site, as previous research suggested [27]. It should be noted, however, that our results did not support a role of wild macaques in the transmission of the disease. The occurrence of Leptospirosis in the wild macaques in this study, however, suggests that pathogenic *leptospira* spp. have persisted as an important bacterial infection in the local wildlife population. Although it seems to be a common contagious bacterial pathogen with subclinical cases in the macaques sampled, its presence

as a wildlife zoonotic disease, which can be transmitted to humans and other animals, presents a concern for both veterinary and public health.

The investigation of pathogenic *Leptospira* bacteria via PCR assay in the blood samples of subclinical long-tailed macaques explained their active infection status and were used to assess the current infection rates within their population, which were determined as potential reservoirs of leptospirosis in this area [28]. However, all of the blood samples were negative for *Leptospira*. This finding may be the result of the sample-collection period occurring during a time phase without asymptomatic bacteremia status of the long-tailed macaques. As such, the presence of bacteria in the blood was not detected [3]. However, the PCR results could not be used to conclude that the leptospirosis was not present in these wild macaques. Further, there is a possible explanation for negative PCR results. Research has shown that the incidence of leptospirosis shows seasonal variation, with cases of the disease increasing with rainfall intensity [29]. The lack of positive PCR results in the current study may be due to the fact that blood sampling was not conducted during the rainy season.

## 5. Conclusions

The purpose of the study was to assess the prevalence of *Leptospira* spp. serovar circulating in a population of free-ranging long-tailed macaques at Kosumpee Forest Park (KFP), Maha Sarakham, Northeastern Thailand. The results revealed the occurrence of leptospirosis in the macaques, which live in close proximity to the local community and have frequent contact with residents and tourists. The findings are crucial in terms of recognizing and identifying the dominant contagious serovars in the study area. Additionally, this evidence-based research on *Leptospira* serovars indicates that several pathogenic *Leptospira* serovars are circulating in the wild macaques in the study and might be associated with previous epidemiological data in both humans and animals in the area. As such, we emphasize the need for concern regarding leptospirosis as a potential zoonotic disease in this area. Additional research is needed to monitor the presence of pathogens in other potential animal reservoirs that could be act as leptospirosis carriers so to improve our understanding of the disease circulation in every part in the ecosystem including wildlife, domestic animals, and humans.

**Author Contributions:** Conceptualization, N.P.; project administration, R.C.K., A.K. and T.T.; sample collection, N.P., P.T., T.T. and A.K.; methodology, N.P., P.T. and T.T.; data analyzing, N.P. and R.C.K.; Drafted and revised the manuscript, N.P., P.K. and R.C.K. All authors have read and agreed to the published version of the manuscript.

**Funding:** This research project was financially supported by Mahasarakham University (Fast Track 2021). R.C.K. and P.K.'s effort was supported in part by the National Institutes of Health (NIH) Office of Research Infrastructure Programs (ORIP) under award number P51OD010425 to the Washington National Primate Research Center, USA.

**Institutional Review Board Statement:** Not applicable.

**Informed Consent Statement:** Not applicable.

**Data Availability Statement:** Not applicable.

**Acknowledgments:** The authors would like to thank all the staff at Kosumpee Forest Park; Pitchakorn Petcharat from Petcharat Animal Clinic; Thanyaphorn Chamnandee, Kittisak Saengthong, Suvit Pathomthanasarn, Ratchanon Kusolsongkhrokul, Papichchaya Doemlim, and Panitporn Damrong-sukij from One Health Research Unit VET MSU; the Wildlife and Exotic Friends Club—MSU; and students from the Faculty of Environment and Resource Studies at MSU for their valuable advice and assistance with the sampling in the field. We are grateful to the Faculty of Veterinary Sciences at MSU for the use of its diagnostic laboratory. Finally, we are grateful to the Thailand Department of National Parks—Wildlife and Plant Conservation (DNP) and the National Research Council of Thailand for their approval (NRCT project approval to RCK—Project ID: 2016/048; "Healthy Coexistence between Human and Non-human Primates: A One Health Approach").

**Conflicts of Interest:** The authors declare that they have no competing interests in this research.

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
