# Peer review of "Leptospira Seroprevalence in Free-Ranging Long-Tailed Macaques (Macaca fascicularis) at Kosumpee Forest Park, Maha Sarakham, Thailand"

_2036-7449, doi:10.3390/idr15010002_

Round 1

Reviewer 1 Report

Indeed, this study shows that macaques at Kosumpee Forest Park may be one of the sources of leptospiral infection in Thailand. The negative PCR result for the LipL32 gene is not surprising. This may be due to the fact that there is no active infection and therefore the sampling time was not the period of bacteremia. In this case, urine samples could be a better material for PCR.

Minor corrections to the text are needed (e.g.: lane 60;  "t" is to be removed); Table 1 - it would be worth adding the serum dilution in MAT (for positive results).

Author Response

Infectious Disease Reports

----------------------------------------------------------------------------------------

Title: Leptospira seroprevalence in free-ranging long-tailed macaques (Macaca fascicularis) at Kosumpee Forest Park, Maha Sarakham, Thailand

Manuscript Number: idr-2027686

Author: Natapol Pumipuntu, Tawatchai Tanee, Pensri Kyes, Penkhae Thamsenanupap, Apichat Karaket, Randall C. Kyes

-------------------------------------------------------------------------------------------------------------------------

Response to reviewers

Reviewer # 1 comments:

Comment 1: Indeed, this study shows that macaques at Kosumpee Forest Park may be one of the sources of leptospiral infection in Thailand. The negative PCR result for the LipL32 gene is not surprising. This may be due to the fact that there is no active infection and therefore the sampling time was not the period of bacteremia. In this case, urine samples could be a better material for PCR.

Authors respond: We accept and followed the reviewer’s comment. Urine samples could be a better material for PCR, but it’s very difficult to collect urine samples from wild macaques.  As such, this is a limitation of our study.

Comment 2: Minor corrections to the text are needed (e.g.: lane 60; "t" is to be removed); Table 1 - it would be worth adding the serum dilution in MAT (for positive results).

Authors respond: Thank you, we have accepted and followed the reviewer’s comment.

On behalf of all authors, I would like to thank all the reviewers very much for your time, your suggestions and comments that have led to much improvement in the quality of our manuscript.

My best regards.

Yours sincerely,

Natapol Pumipuntu, DVM, Ph.D.

7/12/2022

Reviewer 2 Report

The work performed by Pumipuntu et al. provides interesting evidence on the circulation of Leptospira in free-living primates. It provides evidence of Leptospira exposure in long-tailed free ranging macaques living at Kosumpee Forest Park, Maha Sarak- 3 ham, Thailand. Authors detected the presence of anti-leptospira antibodies in three out of thirty individuals by using microscopic agglutination test (MAT). All blood samples were also tested for the presence of Leptospira DNA by using a conventional PCR that amplifyes a fragment of the LipL32 gene, but no positive results were found.

Although the sample size (n=30) is small with respect to the population of macaques living in the park (apron N=850), these results provide important information about how exposed are these animals to leptospirosis. Research on leptospirosis in wild animals that live near domestic animals and humans are important to understand the complexity of the epidemiology of this disease in places with high mammalian diversity. 

I only have concerns about the authors' conclusion in defining macaques as natural reservoirs of pathogenic Leptospira species. The results do not provide evidence that the animals are excreting the bacteria in their urine and therefore this conclusion is unsubstantiated. They also focus the manuscript on the fact that the proximity of humans to these animals could be a risk to human health. This could be dangerous in terms of conservation for these animals, as it could lead to negative reactions from the human populations living around the park.  Therefore, I suggest reviewing the direction in which the results are being approached.

Author Response

Infectious Disease Reports

----------------------------------------------------------------------------------------

Title: Leptospira seroprevalence in free-ranging long-tailed macaques (Macaca fascicularis) at Kosumpee Forest Park, Maha Sarakham, Thailand

Manuscript Number: idr-2027686

Author: Natapol Pumipuntu, Tawatchai Tanee, Pensri Kyes, Penkhae Thamsenanupap, Apichat Karaket, Randall C. Kyes

-------------------------------------------------------------------------------------------------------------------------

Response to reviewers

Reviewer # 2 comments:

The work performed by Pumipuntu et al. provides interesting evidence on the circulation of Leptospira in free-living primates. It provides evidence of Leptospira exposure in long-tailed free ranging macaques living at Kosumpee Forest Park, Maha Sarak- 3 ham, Thailand. Authors detected the presence of anti-leptospira antibodies in three out of thirty individuals by using microscopic agglutination test (MAT). All blood samples were also tested for the presence of Leptospira DNA by using a conventional PCR that amplifyes a fragment of the LipL32 gene, but no positive results were found.

Although the sample size (n=30) is small with respect to the population of macaques living in the park (apron N=850), these results provide important information about how exposed are these animals to leptospirosis. Research on leptospirosis in wild animals that live near domestic animals and humans are important to understand the complexity of the epidemiology of this disease in places with high mammalian diversity.

I only have concerns about the authors' conclusion in defining macaques as natural reservoirs of pathogenic Leptospira species. The results do not provide evidence that the animals are excreting the bacteria in their urine and therefore this conclusion is unsubstantiated. They also focus the manuscript on the fact that the proximity of humans to these animals could be a risk to human health. This could be dangerous in terms of conservation for these animals, as it could lead to negative reactions from the human populations living around the park. Therefore, I suggest reviewing the direction in which the results are being approached. Additionally, there are some methodological issues that are review below. Please check typos, spelling, and phrasing along the entire document.

Authors respond: Thank you for your thoughtful suggestion. We have accepted and followed the reviewer’s comment.

Comment 1:

Introduction:

  1. Please check the first paragraph structure and writing.

Authors respond: We have accepted and followed the reviewer’s comment.

  1. Authors cite reports of leptospirosis in domestic animals, what about wild live? What is known about Leptospirosis in free ranging primates? The article is about a wildlife primate specie, making this point relevant.

Authors respond: We have accepted the reviewer’s comment. From our knowledge, the study of leptospirosis infection in wildlife in Thailand is quite limited because the difficulty of sample collection and the related restrictions from wildlife conservation and protection acts in Thailand. However, some recent studies of leptospirosis infection in macaques from Thailand and other countries have been conducted and we detailed them in the discussion section.

Comment 2:

Methods:

  1. Figure 1. Please add the word “Thailand” on the map that is indicating the specific location of the park where the study took place.

Authors respond: We have accepted the reviewer’s comment. We added the word “Thailand” on the map in Figure 1 in a revised file.

  1. Line 189, cite reference 27, please check that reference correspond to the cite. In this case, it is talking about leptospirosis in Thailand and it is refereeing to COVID 19! Please check all references and citations.
    Authors respond: Thank you noting this. We have accepted the reviewer’s comment. We changed the reference number 27 from Breedon et al., 2021 to Toyokawa et al., 2011 as “Toyokawa, T.; Ohnishi, M.; Koizumi, N. Diagnosis of acute leptospirosis. Expert Rev. Anti. Infect. Ther. 2011, 9, 111-121.” in the references part. We also checked all references and citations in the revised file.

  2. Line 194-198, “... a possible explanation for the negative PCR … due to the fact that blood sampling was not conducted during rainy season”. This might be a limitation of the study?
    Authors respond: This is also a limitation of the study. Given that the study site is located near the Chi-river, it experiences frequent flooding every year during the rainy season. As such, we could not get into the forest park to collect samples until the water level h decreased.

  1. Why were you looking for leptospira in blood? When the bacteria is found in blood it highly suggest acute infection, and You would expect leptospira in blood only in the first 7 to 10 days after infection. What are you aiming by looking for leptospira DNA in blood?
    Authors respond: The detection of leptospiral DNA by PCR in this study was applied for early diagnosis of leptospirosis in macaques for the reason that the prompt diagnosis of Leptospirosis via PCR is essential for the management of both animal cases as well as for the park staff, and the efficient implementation of animal health measures. In addition, we believe that the use of PCR with MAT in the early phase and chronic phase of leptospirosis, respectively, can help increase the detection rate.

  2. What about taking urine samples as evidence of shedding?
    Authors respond: The detection of leptospirosis in urine sample is also useful in studying long-term shedding. However, the urine sample collection method for wildlife is also a limitation of the study. The urine sampling can be invasive for wildlife and sufficient amounts of urine from wild monkeys is very difficult to collect.

  3. Reference 20 is missing its number; other references have duplicated numbers (ej. 21 and 22).
    Authors respond: Thank you for your notice, reference number 20 appears in the Discussion part at line 162.

  1. Authors cite Stoddard et al. for the Lipl32 PCR, however they describe a conventional PCR that is visualized on an agar gel after electrophoresis. Stoddard et al. assay is standardized as a Taqman PCR. Therefore, the results of the PCR are not reliable.
    Authors respond: Thank you for your comment. We rechecked the reference number 19 and adjusted it from Stoddard et al., 2009 to Tan et al., 2014 as Tan, C.G.; Dharmarajan, G.; Beasley, J.; Rhodes, O. Jr.; Moore, G.; Wu, C.C.; Lin, T.L. Neglected leptospirosis in raccoons (Procyon lotor) in Indiana, USA. Vet. Q. 2014, 34, 1-10.

Comment 3:

Results:

  1. Table 1, please include the antibody titers of MAT (1:100? 1:1600?). Please discuss these results in the corresponding section.

Authors respond:  We have accepted and followed the reviewer’s comment.

Comment 3:

Discussion and Conclusions:

  1. Please discuss and cite about human cases in the area that suggest you that these animals are in fact natural reservoirs of leptospirosis.

Authors respond: For the human cases of leptospirosis in Kosum Phisai district (where the KFP located), from 2004-2014 the district had the second largest number of cases in Maha Sarakham province. Interestingly, various types of animal such as buffaloes, cattle and dogs in this area may play a potential role as important leptospirosis animal reservoirs at this study site (as mentioned in a previous study by Viroj et al., 2021 that we have cited in the revised manuscript). Many serovars found in those animals can be pathogenic in humans. However, they have no previous study of leptospirosis in the wild macaques at this study site. By the way, our results did not support a role for wild macaques in the transmission of the disease, but they could play a potential role as natural reservoirs of leptospirosis. We added this discussion at line 179-185 in the revised manuscript.

  1. Authors mention that there are interactions between macaques and humans that can be a potential risk for infection. Could you mention which specific behaviors or interactions?

Authors respond: The macaques in this study area have adapted to live in a share space with the human community because of a loss of the natural habitat and lack of natural foodstuff. As such, there is frequent conflict between the local residents and macaques, such as the macaques invading and destroying  agricultural areas, property and housing. in turn, the local people are likely at higher risk of pathogen transmission and  contracting disease caused by the macaques. The tourists also have a potential risk for infection by direct interaction with the monkeys such as hand feeding them fruits or beans without gloves or any PPEs, and indirect contact with  surfaces, food or water contaminated with pathogens shed  by the macaques. 

  1. How possible is that domestic animals are responsible for macaques’ infection?

Authors respond: The area of Kosumpee Forest Park is located adjacent to the human community and agricultural areas as shown on the map in Figure 1. Therefore, there is a coexistence among livestock, companion animals and the wild macaques. Moreover, there is a frequent presence of domestic animals such as dogs and cattle inside and around the forest park areas. This suggests that it might be possible for the macaques to acquire pathogens, including leptospiral bacteria, from other animals and vice versa. However, further work is neededto address the direction or tracking of leptospirosis transmission, perhaps via pathogen genetic identification.

  1. Macaques are natural reservoirs? Or have been exposed to Leptospirosis? There is no evidence of shedding in urine.

Authors respond: From our results, we can only determine that the macaques have been exposed to leptospirosis, and there is the possibility that they might  play a role as natural reservoirs. Further studies are needed to obtain evidence of pathogen shedding from the macaques. Thank you for your comment.  

On behalf of all authors, I would like to thank all the reviewers very much for your time, your suggestions and comments that have led to much improvement in the quality of our manuscript.

My best regards.

Yours sincerely,

Natapol Pumipuntu, DVM, Ph.D.

7/12/2022
